# High-Fat Diet-Induced Obesity Increases Brain Mitochondrial Complex I and Lipoxidation-Derived Protein Damage

**DOI:** 10.3390/antiox13020161

**Published:** 2024-01-26

**Authors:** Rebeca Berdún, Èlia Obis, Natàlia Mota-Martorell, Anna Bassols, Daniel Valent, José C. E. Serrano, Meritxell Martín-Garí, María Rodríguez-Palmero, José Antonio Moreno-Muñoz, Joan Tibau, Raquel Quintanilla, Reinald Pamplona, Manuel Portero-Otín, Mariona Jové

**Affiliations:** 1Department of Experimental Medicine, Lleida Biomedical Research Institute (IRBLleida), University of Lleida (UdL), 25198 Lleida, Spain; rebecaberdun@gmail.com (R.B.); eobis@irblleida.cat (È.O.); natalia.mota@udl.cat (N.M.-M.); josecarlos.serrano@udl.cat (J.C.E.S.); mmartin@irblleida.cat (M.M.-G.); reinald.pamplona@udl.cat (R.P.); 2Departament de Bioquímica i Biologia Molecular, Facultat de Veterinària, Universitat Autònoma de Barcelona, 08025 Barcelona, Spain; anna.bassols@uab.cat (A.B.); daniel.valent@uab.cat (D.V.); 3Laboratorios Ordesa S.L., Barcelona Science Park, 08028 Barcelona, Spain; maria.rodriguez@ordesa.es (M.R.-P.); josea.moreno@ordesalab.com (J.A.M.-M.); 4Animal Science—Institut de Recerca i Tecnologia Agroalimentàries, IRTA, Monells, 17121 Girona, Spain; joan.tibau@irta.cat; 5Animal Breeding and Genetics Program, IRTA, Torre Marimon, 08140 Caldes de Montbui, Spain; raquel.quintanilla@irta.cat

**Keywords:** obesity, lipoxidation, mitochondrial complexes, n3 PUFA, probiotics, age-related neurodegenerative diseases

## Abstract

Obesity is a risk factor for highly prevalent age-related neurodegenerative diseases, the pathogenesis of whichinvolves mitochondrial dysfunction and protein oxidative damage. Lipoxidation, driven by high levels of peroxidizable unsaturated fatty acids and low antioxidant protection of the brain, stands out as a significant risk factor. To gain information on the relationship between obesity and brain molecular damage, in a porcine model of obesity we evaluated (1) the level of mitochondrial respiratory chain complexes, as the main source of free radical generation, by Western blot; (2) the fatty acid profile by gas chromatography; and (3) the oxidative modification of proteins by mass spectrometry. The results demonstrate a selectively higher amount of the lipoxidation-derived biomarker malondialdehyde-lysine (MDAL) (34% increase) in the frontal cortex, and positive correlations between MDAL and LDL levels and body weight. No changes were observed in brain fatty acid profile by the high-fat diet, and the increased lipid peroxidative modification was associated with increased levels of mitochondrial complex I (NDUFS3 and NDUFA9 subunits) and complex II (flavoprotein). Interestingly, introducing n3 fatty acids and a probiotic in the high-fat diet prevented the observed changes, suggesting that dietary components can modulate protein oxidative modification at the cerebral level and opening new possibilities in neurodegenerative diseases’ prevention.

## 1. Introduction

Obesity is a risk factor for age-related diseases. For instance, obesity has been previously associated with age-related neurodegenerative processes, such as mild cognitive impairment and Alzheimer’s disease (AD) [1,2,3,4]. Insulin receptors are present in several neural cell types, including glia and neurons [5], suggesting that AD might be related to insulin resistance [6,7]. Obesity-induced insulin resistance could fuel some of the culprits of age-related neurodegeneration, such as mitochondrial dysfunction, amyloid-beta deposit, protein aggregation, and neuroinflammation, to name a few. Furthermore, since brain vasculature is essential for neural tissue homeostasis, the obesity–neurodegeneration link has also been attributed to changes in vessel function [8]. Finally, this link might also comprise the influence of adipose tissue-derived products (i.e., inflammatory mediators) and free fatty acids (FFA), and oxidative stress status.

Circulating FFAs are generated due to alterations in expanding adipose tissue and have a lipotoxic role in both peripheral and nervous tissue. FFAs can target the central nervous system (CNS), which, together with the chronic low-grade inflammatory state, would result in cognitive impairment, other CNS diseases, and peripheral neuropathies [9]. To model this increased risk of neurological complications, the available experimental models of obesity have some intrinsic limitations. Increased obesity is observed in rodents when placed on a higher-fat diet. However, due to physiological differences, specific cardiometabolic markers (such as low HDL cholesterol, increased LDL cholesterol, and coronary calcification) translate better in pigs and primates than in rodent models [10,11]. In fact, lipoprotein metabolism in primates and swine is very similar to that of humans. Therefore, unhealthy diets in these species can induce changes in lipid profiles associated with cardiometabolic risk, reproducing many changes in humans. Pigs are particularly suited for evaluating lipid metabolism based on the knowledge of the genetic traits controlling it and the possibility of monitoring the buildup of atherosclerosis by using clinical imaging systems such as computed tomography. However, a detailed analysis of the potential effects of obesity on the brain is not available.

Oxidative stress comprises the pathophysiological mechanisms behind age-related neuronal impairment [12,13,14]. Typically, protein oxidative damage in brain age-related diseases has been evaluated by measuring protein carbonyls derived from the direct interaction of reactive oxygen species with susceptible amino acid residues. However, third-party molecules are also involved in the chemical pathways connecting enhanced free radical efflux and protein structural alteration, which could increase protein damage [15,16]. Cellular processes such as glycolysis and triose phosphate metabolism [16] could contribute to the production of these intermediate compounds that interact with lysine, arginine, and cysteine residues in proteins, leading to the formation of stable advanced glycation end-products (AGEs). Polyunsaturated fatty acids (PUFA) are another group of third-party compounds highly vulnerable to oxidation that produce a wide variety of reactive aldehydes [17,18]. These, in turn, are capable of non-enzymatically reacting with proteins, leading to the formation of advanced lipoxidation end-products (ALEs). The enriched PUFA content of the brain and its high oxygen consumption provide evidence for the potential importance of lipid peroxidation-derived mechanisms in the etiology of brain aging and AD [19]. Reinforcing this idea, immunohistochemistry reveals non-enzymatic protein damage in AD lesions [20,21,22], which contrasts with the lack of studies evaluating the effect of diet-induced obesity on brain protein damage.

In the present work, we have used a porcine model to evaluate the effects of high-fat diet-induced obesity on the content of oxidative modification markers in brain cortex proteins. Since these are related to free radical production and lipid composition, we have also measured those parameters by evaluating the content of mitochondrial respiratory complexes and the fatty acid (FA) profile in brain tissue. The results reveal a specific profile of molecular modification in high-fat diets in female piglets, with a dominance of lipid peroxidative damage (34% increase in malondialdehyde-lysine (MDAL)), without changes in the FA profiles, and with specific changes in respiratory mitochondrial complex I (subunits NDUFS3 and NDUFA9) and complex II (flavoprotein, FP). To gain information on the dynamic behavior of the system, we compared these effects with those of a high-fat diet enriched with a probiotic and the same probiotic with n3 PUFA, with properties against high-fat metabolic disorders [23,24,25]. The results demonstrate that the degree of protein lipoxidation depends not only on calories but also on the dietary components, demonstrating the protecting role of n3 PUFA. These findings pave the way for exploring the potential involvement of n3 PUFA and probiotics in preventing neurodegenerative pathologies.

## 2. Materials and Methods

### 2.1. Design and Ethics

The ARRIVE guidelines in the Essential 10 set have been followed (https://arriveguidelines.org/arrive-guidelines, accessed on 26 January 2024). All the experimental procedures, including management, trait recording, and monitoring, were approved by the Ethical Committee of the Institut de Recerca i Tecnologia Agroalimentàries (IRTA, Girona, Spain) (DAAM 7306/2013) and were performed according to the Spanish Policy for Animal Protection RD1201/05, which meets European Union Directive 86/609 about the protection of animals used in experimentation.

A total of 43 female piglets from a Duroc pig line (Sus scrofa domesticus) were used in the present study. Animals were born in 11 different litters (i.e., 11 groups of 3–4 littermates). Same-sex littermates, from the same father and mother, were randomly distributed into 4 experimental groups using a matched-pairs experimental design. After weaning, piglets were transferred to the IRTA pig experimental station and subjected to the same management procedures as described [23,26]. Briefly, at 9 weeks of age, animals were located in environmentally monitored facilities, randomly distributed into 4 pens (10–11 animals per pen from different litters), and fed ad libitum for 10 weeks with 4 different dietary treatments, giving rise to four different experimental groups (see below). Each pen had a partly slatted floor (60% solid concrete and 40% slatted), with some sawdust provided on the concrete floor on a regular basis and a natural light cycle with a minimum of intensity of 40 lx (EU legislation on pig welfare) and 8 h light. The room temperature was maintained at 22 ± 5 °C.

At nine weeks of age, animals were divided into four food regimens (ten to eleven animals per treatment) ad libitum for ten weeks: a standard (and balanced) growth diet under the NRC’s (Nutrition Resource Centre) suggestions (T1); a Western-style diet with a high fat content and protein from animal sources (caseinate (T2)); the same Western-style diet with a 50% replacement of protein by protein from vegetable sources (rice hydrolysate), containing 5050 cfu/day of the patented strain of *Bifidobacterium breve* (CECT8242) (T3); and T3 with n3 PUFA added to the diet (1 g of stearidonic acid and 2 g of docosahexaenoic acid per 100 g of fat) (T4). Feeds were prepared at the Mas de Bover IRTA center (Tarragona, Spain). Detailed information about their components and nutritional composition is provided elsewhere [26].

All pigs were individually weighed at the experiment’s start, every two weeks throughout, and the day before slaughter. Electronic feeders (IVO-station feeders; INSENTEC^®^, Marknesse, The Netherlands) placed in each pen allowed for recording the individual feed intake. All pigs were scanned using computed tomography (CT) at around 18 weeks to produce one axial image at the level of the second lumbar vertebrae to evaluate adipose tissue distribution. Pigs were CT-scanned using General Electric HiSpeed ZX/I (General Electric, Fairfield, CN, USA) equipment while under anesthesia and after a 16 h fast. VisualPork was used for the image analyses [26] to convert the areas of interest into volumes (mm^3^) of various fat depots, resulting in estimations of the relative fat volume and the relative amount of subcutaneous, intermuscular, and flare fat.

Animals were killed at the IRTA experimental slaughterhouse when they reached the age of 19 weeks under completely controlled circumstances and per all welfare laws. Animals were fasted 8 h before being transported from the experimental farm to the experimental slaughterhouse (1.2 km of distance). After the unloading, pigs were located in the lairage pens for an hour. Animals were stunned in groups of two by exposure to 90% CO_2_ in atmospheric air for 3 min and exsanguinated afterwards. The brain was removed from the skull immediately after slaughter. Meningeal tissues and vessels were eliminated, and prefrontal cortex (PFC) samples were obtained as indicated [24] and frozen at −80 °C. Serum biochemical analyses were obtained as detailed previously [26].

### 2.2. Protein Non-Enzymatic Modification Markers

Markers of protein oxidation (the protein carbonyl glutamic semialdehyde [GSA]), glycoxidation (Nε-(carboxyethyl)-lysine [CEL], Nε-(carboxymethyl)-lysine [CML], and carboxymethyl-cysteine [CMC]), and lipoxidation (MDAL) were determined as trifluoroacetic acid methyl esters (TFAME) derivatives in acid-hydrolyzed delipidated and reduced proteins from PFC samples by gas chromatography/mass spectrometry (GC/MS) as detailed in [19,27]. Briefly, cortical brain tissue samples were homogenized at a ratio of 1:20 (*w*/*v*) in a buffer containing 180 mM KCl, 5 mM MOPS, 2 mM EDTA, 1 mM diethylenetriaminepentaacetic acid, and 1μM butylated hydroxytoluene to avoid the artifactual formation of protein carbonyl [19], along with 10 µg/mL aprotinin and 1 mM phenylmethylsulfonyl fluoride, maintaining a pH of 7.4 using a Potter-Elvehjem device at 4 °C. Following centrifugation (500× *g* for 5 min) to pellet unbroken tissue, supernatants containing 0.75–1 mg of protein were subjected to delipidation through chloroform/methanol (2:1, *v*/*v*). The lipids were reserved for FA analysis, and the proteins were precipitated using 10% trichloroacetic acid and subsequent centrifugation steps. Subsequently, proteins underwent an overnight reduction in a 0.5 M NaBH4 solution (final concentration) in a pH 9.2 borate buffer supplemented with one drop of hexanol as an anti-foam agent. The proteins were re-precipitated using 10% trichloroacetic acid and centrifuged. For relative quantification, known quantities of [2H8]lysine (d8-Lys); C, [2H2]CML (d2-CML), [2H4]CEL (d4-CEL), [2H8]MDAL (d8-MDAL), [2H5]5-HAVA (stable derivatives of GSA), and [13C3-15N] CMC were included as internal standards. Protein hydrolysis was achieved through incubation at 155 °C for 30 min in 1 mL of 6 N HCl. Following drying, derivatization involved dissolving hydrolysates in 1.5 mL of freshly prepared 1N methanolic HCl, heated at 65 °C for 30 min. After solvent evaporation, 1.5 mL of trifluoroacetic anhydride was added, leading to a mixture incubating at room temperature for 1 h. The resulting N, O-trifluoroacetyl methyl ester derivatives were subjected to analysis using an Agilent model 6890 gas chromatograph equipped with a 30 m HP-5MS capillary column (30 m × 0.25 mm × 0.25 µm) coupled with an Agilent model 5973A mass selective detector. Analyses were carried out by selected ion-monitoring GC/MS (SIM-GC/MS). The ions used were: lysine and [2H8]lysine, *m*/*z* 180 and 187, respectively; 5-hydroxy-2-aminovaleric acid (HAVA, stable derivatives of GSA), *m*/*z* 294 and 298, respectively; CML and [2H2]CML, *m*/*z* 392 and 394, respectively; CEL and [2H4]CEL, *m*/*z* 379 and 383, CMC and [13C3-15N]CMC, *m*/*z* 273 and 275, respectively; and MDAL and [2H8]MDAL, *m*/*z* 474 and 482, respectively. The amounts of product were expressed as μmoles of GSA, CML, CEL, CMC, or MDAL per mol of lysine.

### 2.3. Mitochondrial Electron Transport Chain Complexes

The content of the different mitochondrial respiratory chain complexes in pig brain cortex was analyzed through Western blot analyses of specific peptides, following established protocols [28,29]. In brief, homogenates (20 micrograms of protein) of the PFC (prepared as previously outlined) were redissolved in a buffer containing 62.5 mM Tris-HCl at pH 6.8, 2% SDS, 10% glycerol, 20% β-mercaptoethanol, and 0.02% bromophenol blue, and subjected to a 3 min heating at 95 °C. Subsequently, the protein samples underwent one-dimensional electrophoresis utilizing SDS and were transferred onto PVDF membranes. These membranes were immersed in a blocking solution comprising 2 M Tris, 2.5 M NaCl, 5% BSA, and 0.01% Tween at room temperature for 1 h. Immunoassays were conducted using antibodies targeting specific proteins, namely NDUFV2, NDUFS3, NDUFS4, NDUFS5, NDUFA9, FP (also called succinate dehydrogenase complex FP subunit A (SDHA)), CORE2 (UQCRC2), COX1, and VDAC-1/Porin. The secondary antibodies employed included anti-mouse and anti-rabbit antibodies. The resultant protein bands were visualized using an enhanced chemiluminescence HRP substrate (Millipore, MA, USA). Signal quantification and capture were accomplished using ChemiDoc equipment (Bio-Rad Laboratories, Inc., Barcelona, Spain). Protein content was derived by computing the ratio of densitometry values related to their respective Coomassie staining. The absence of protein staining was confirmed when primary antibodies were excluded, verifying antibody specificity. Images of raw Western blots are shown in Appendix A.

### 2.4. FA Profile of the PFC Lipidome

FA profiling of the lipidome from brain cortex samples was performed following established procedures [19,30]. Briefly, after lipid extraction, the organic (chloroform) fraction was subjected to transesterification using 5% methanolic HCl in a 2 mL solution and heated at 75 °C for 90 min. The resulting FA methyl esters were extracted using 2 mL of n-pentane and 1 mL of saturated NaCl solution, evaporated using N_2_ gas, and eventually dissolved in 80 µL of carbon disulfide. The subsequent analysis involved gas chromatography, as described in previous literature [31], identifying FA methyl esters achieved by comparison with authenticated standards. FA amounts were quantified as a percentage relative to the complete chromatogram, and FA indices, and the calculated activity of elongases and desaturases were determined using the previously described methods [30].

### 2.5. Statistics

Prism 10 for Windows 64b (v 10.1.2 (324 (GraphPad, Boston, MA, USA)) and the SPSS Statistics 25.0 (IBM, NYC, NY) packages were used for data analysis and graph production. Weight gain and biochemical and protein oxidative modifications were analyzed by Brown–Forsythe and Welch one-way ANOVA tests—assuming that variances differed between groups—employing Dunnett T3 correction for multiple comparison using statistical hypothesis testing for post hoc analyses, except for fatty acid derived indexes, where we employed the two-stage linear step-up procedure of Benjamini, Krieger, and Yekutieli for controlling false-discovery rate. The Spearman’s correlation coefficient rho was used to analyze the correlation between variables. Samples that did not meet adequate quality standards, either due to insufficient material or necropsy failures, were excluded from our analyses. All remaining samples were analyzed individually, except in the case of Western blot analyses, where we pooled all samples with a minimum amount and quality belonging to the same experimental group (10 samples for T1, 8 samples for T2, 7 samples for T3, and 12 samples for T4) and performed two to three technical replicates. Investigators were blinded to the treatments. Significance was considered when the *p* value < 0.05.

## 3. Results

### 3.1. Biochemical and Phenotipic Differences between Pigs Fed Different Diets

Body weight and serum lipid variables of the different experimental groups are shown in Table 1. Consistent with previous findings [26], animals fed a T2 diet exhibited increased body weight, elevated circulating triacylglyceride levels, and elevated LDL and HDL-cholesterol concentrations. Regimens T3 and T4 abrogated the weight gain induced by T2, but not all serum lipid measurements were normalized.

### 3.2. A Western-Type Diet Increases Protein Lipoxidative Modifications on the Brain Cortex

The effects of a Western-type diet (T2) on protein oxidative damage markers in the pig brain were assessed (Figure 1). The results demonstrated that after 10 weeks of high-fat diet consumption, only lipoxidation increased in PFC (MDAL, *p* = 0.0156) (Figure 1C). Protein carbonyl (GSA) and non-enzymatic protein modifications derived from glycoxidation reactions (CML, CEL, and CMC) were not influenced by the T2 diet (Figure 1A,B), suggesting a selective impact on lipid oxidation. As reported for the human brain [19], the steady-state levels of protein carbonylation (GSA) were the highest, making protein carbonylation the most common protein modification in the brain (Figure 1A).

### 3.3. The Brain Cortex Protein Lipoxidative Modifications Induced by the Western Diet Are Not Due to Changes in FA Composition

The FA composition of the brain’s lipidome was determined (Table 2). The results indicate that a high-fat regimen (T2) does not modify the brain PFA FA profile.

### 3.4. Western-Type Diet Is Associated with Changes in the Content of Mitochondrial Respiratory Complexes

The content of the mitochondrial respiratory chain in the brain was evaluated (Figure 2). To estimate whether dietary regimes caused any change in mitochondrial mass, porin levels were quantified. Therefore, all measured mitochondrial respiratory complex peptides were adjusted to porin levels. The high fat intake (T2) infringed a significant increase in NDUFS3 and NDUFA9 complex I subunits (*p* < 0.01 and 0.049, Figure 2A) and the FP of complex II (*p* = 0.026, Figure 2B). Reinforcing the specificity of this phenomenon, the subunits of complexes III and IV studied were not affected by the T2 diet (Figure 2C,D).

### 3.5. Dietary Supplementation with Vegetal Protein with Bifidobacterium Breve CECT8242 Alone or in Combination with n3 Fatty Acids Can Partially Revert the Effects of a Western-Type Diet in the Brain Cortex Protein Lipoxidative Damage

To evaluate dietary modifiers of high-fat consequences, we measured the above-mentioned parameters in animals under a T2 diet containing vegetal protein supplemented with *Bifidobacterium breve* CECT8242 alone (T3) or in combination with n3 PUFA (T4). The biochemical and morphological traits demonstrated that the T4 diet reduces the levels of HDL-cholesterol and triglycerides when compared with T3, suggesting that n3 supplementation is responsible for the effects observed in these parameters (Table 1).

Oxidative damage derived from the Western diet (T2) in the brain was reversed after following a T4 diet. In fact, the results showed that n3 supplementation (T4) restored lipoxidation levels to those of a conventional diet (MDAL, *p* = 0.0191) (Figure 1C). Additionally, T3 reduced protein carbonylation (GSA, *p* = 0.014) (Figure 1A).

Furthermore, supporting the role of mitochondrial changes after MDAL accumulation, the increased content of complex I subunits NDUFS3 and NDUFA9 after high fat intake (T2) was reversed by the T3 and T4 diets (Figure 2A), an effect that seems to be caused by the probiotic administration and potentiated by n3 supplementation. In addition, n3 dietary supplementation reduced NDUFS4 complex I subunits (Figure 2A).

Finally, the effect of the T3 and T4 diets on the brain FA profile was determined (Table 2). The results revealed that dietary n3 supplementation (T4) modifies FA lipid composition. Specifically, T4 lowered C18:1n9, C22:5n6, and C22:5n3 levels and increased C18:3n3, C20:3n6, C20:5n3, and C22:6n3 levels. Globally, all of these changes resulted in increased PUFA levels in PFC after n3 supplementation (T4), as well as a higher double bound, peroxidability, and anti-inflammatory indexes.

### 3.6. Lipoxidation-Derived Protein Damage in the Brain Cortex Correlates with Peripheral Lipids

Finally, we performed correlation analyses between protein oxidative modification and peripheral traits of lipid metabolism. We found positive correlations between the blood LDL levels and the weight of the animals with the brain lipoxidative damage biomarker MDAL (Figure 3).

## 4. Discussion

Our results demonstrate that a Western-type diet applied to female pigs increases brain lipoxidative damage, an effect that can be reversed by applying a probiotic n3-supplemented diet. Numerous studies have indicated a correlation between obesity and the risk of developing AD [1,4,32,33]. This risk represents a modifiable factor in neurodegenerative processes [1,4], suggesting that dietary components could serve as preventive measures. Several explanations for the link existing between obesity and AD have been proposed [34,35]. One possibility is that obesity enhances inflammation in the brain, which can damage neurons and promote the development of AD [36]. Another is that obesity is linked to cerebrovascular demise and the accumulation of beta-amyloid, probably through phenomena leading to the loss of grey matter [8].

Several types of oxidative modifications have been found in the brain proteins that diet can influence, including carbonylation, [19] nitration [37], and other oxidative modifications such as direct oxidation, glycoxidation, lipid peroxidation, and DNA damage [38]. Protein carbonylation can lead to changes in protein structure and function and an increased susceptibility to degradation. In line with this, it has been previously described that high-fat diet-induced obesity is associated with increased brain protein carbonylation [39,40]. Protein nitration has been linked to several neurodegenerative diseases. Interestingly, a previous study [41] found that high-fat diet feeding was associated with increased protein nitration in the mouse hypothalamus, accompanied by a lower food intake and higher body weight, suggesting a role in developing metabolic disorders. All of these oxidative modifications can also be influenced by diet and may contribute to the development of neurological and metabolic disorders.

Mitochondrial respiratory chain complexes, particularly complex I, are the primary sources of free radicals [42,43]. Dysregulation of mitochondrial respiratory complexes can lead to decreased ATP production and increased oxidative stress, which can contribute to the development of metabolic and neurological disorders, paving the way for future dementia. In the present study, we demonstrated that a Western-type diet can increase PFC mitochondrial respiratory complexes I (subunits NDUFS3 and NDUFA9) and II levels (FP) in a porcine model of obesity. Interestingly, these effects are reversible by administering a probiotic n3-supplemented diet. In line with this, a previous study investigating the effects of a high-fat diet on mitochondrial respiratory complexes demonstrated that this diet was associated with the altered protein expression of several mitochondrial respiratory complexes in the hypothalamus, including complexes I, III, and V [44]. These results suggest that obesity influences the content and activity of specific mitochondrial respiratory complexes in the brain, which may contribute to the development of metabolic and neurological disorders. Despite the limitations of evaluating mitochondrial mass derived from a single marker (porin), after its quantitation in mitochondrial extracts from PFC samples, no significant differences appeared to be induced by dietary treatments. This would agree with a lack of general effect on mitochondrial biogenesis or in the turnover generated by dietary fat or probiotics. Of note, as VDAC1/porin levels are inversely proportional to lifespan when comparing related species [29], its abundance might be expected to increase under pro-aging conditions such as the high-fat diet to facilitate the transport of hydrogen peroxide from the mitochondrial intermembrane space to the cytoplasm Thus, changes present in specific respiratory complex peptides would suggest functional adaptations to, e.g., supercomplex formation or efficiency [45]. Further research is needed to understand the mechanisms underlying these effects and develop effective strategies for preventing or treating mitochondrial dysfunction in the brain.

Lipid composition, especially peroxidizability, is a significant factor in explaining increased lipoxidative modifications [46]. The results of the present work indicated that the FA profile of the PFC was not affected by the Western-type diet (T2) (Table 1), suggesting that the increased lipoxidative damage observed can be a consequence of structural changes at the mitochondrial level (and, specifically, in the structure of the electron transport chain) rather than the result of changes in membrane FA composition. Indeed, our results indicated that the bioavailability of 22:6n3 in the brain tissue is higher than 18:4n3 because the latter levels do not change with n3 supplementation. The increased levels of C22:6 seem to induce an adaptive response in the n3 biosynthesis pathway, which implies a decreased activity of the elongases Elovl2 and Elovl5 that led to an increase in the C20:5n3 and C18:3n3 PFC levels. Specifically, the changes observed in the n9 and n6 biosynthesis pathways in the present work could be attributed to compensatory mechanisms.

The results presented here also indicate that dietary regimens can alleviate oxidative damage burdens in brain regions. MDAL formation depends on the degree of unsaturation of biological membranes and the intracellular oxidative condition, and its concentration is attenuated by anti-aging nutritional interventions [47]. The fact that the intake of neuroprotective, albeit easily peroxidizable, n3 PUFA (such as that present in the T4 group) was able to prevent or diminish the buildup of MDAL accumulation agrees with a highly dynamic system governing the levels of MDAL in proteins. Previously, n3 PUFA, particularly eicosapentaenoic acid (EPA) and docosahexaenoic acid (DHA), had been shown to benefit mitochondrial function and protein oxidative damage in the brain. Importantly, it has previously been described that dietary PUFA is able to alter brain gene expression, including important genes related to synaptic plasticity and learning in rats [48]. Furthermore, a previous study [49] demonstrated that DHA supplementation was associated with a preventive effect over inflammatory gene expression in the hippocampus in rats without changes in weight gain. Regarding mitochondrial structure and function, DHA is associated with enhanced complex II and complex III enzyme activities in astrocytes [50]. Additionally, n3 PUFA supplementation was associated with changes in mitochondria ADP metabolism ex vivo in human tissues, without major changes in mitochondrial complexes and pointing to relevant roles of membrane phospholipids [51]. In our study, the combination of probiotics with n3 PUFA was able to reverse the effect of a high-fat diet on PFC respiratory complexes. Complementary studies are needed to better understand these diets’ role in the modulation of mitochondrial function.

Emerging theories link mitochondrial dysfunction to neurological and neurodevelopmental disorders, suggesting a causative role beyond correlation [52]. The significance of mitochondria in the gut–brain axis emerges as gut microbiota metabolites impact brain mitochondrial function. The gut microbiome’s potential to impact mitochondrial function through molecules like FA, bile acids, and neurotransmitters is evident. Commensal microorganism metabolites affect brain mitochondria via the blood–brain barrier or vagus nerve, influencing quality, survival, or damage. In this sense, disrupted microbiota–gut–brain communication could disrupt neuronal development. The gut microbiome, influenced by probiotics, involves mitochondria [53,54,55]. Probiotics enhance mitochondrial function dynamics, modulate the microbiota–gut–brain axis and counteract neurodegeneration [52]. In this sense, our results suggest that supplementing a Western diet with probiotics and n3 PUFA may benefit mitochondrial function and protein oxidative damage in the brain and other tissues in an animal model of obesity. Further research is needed to understand the mechanisms underlying these effects and develop effective strategies for preventing or treating mitochondrial dysfunction and oxidative damage in the brain.

Finally, the tendency to increase HDL concentrations, especially in the case of all investigative diets, can be explained by changes in lipid metabolism, previously reported in this model [25], and it could be of interest to interpret the role of lipoproteins as modulating agents in neural homeostasis. In line with this, recent data show that small HDLs are associated with cognitive function in humans, suggesting their protective role [56] and reinforcing the relevant role of neural lipid metabolism in brain homeostasis, recently exemplified by lipidomic studies [57].

The present study has some limitations: (i) the study only includes female pigs, so the effect of these interventions on males has been ignored. This fact could limit the application of these results to humans; (ii) the study is performed on young pigs. This implies, among other things, that it cannot be considered a model of age-related neurodegeneration but rather a model of obesity linked to brain lipoxidative damage. More studies, including those using older pigs, should be performed to evaluate how these interventions modify brain lipoxidative damage in adulthood; (iii) further studies analyzing the specific diet-associated changes in the microbiome are needed to further elucidate the effect of probiotic n3 supplementation on the gut–brain axis; (iv) we have not evaluated actual respiratory complex activities, which would strengthen our conclusions on whether mitochondrial dysfunction could potentially contribute to increased peroxidative damage. Future studies evaluating the effects of a high-fat diet on mitochondrial function in the brain would benefit from the measurement of respiratory complex activity assays; and (v) the lack of perfusion might have some impact on measured markers. However, both the fatty acid composition of blood, which is strikingly different from that of neural tissue, and the low amount of mitochondrial complexes in whole blood in comparison to the tissue suggest that the potential blood contamination would have a low impact.

## 5. Conclusions

Globally, the present study demonstrated that the effects of a Western-type diet on the brain’s lipoxidative status could be partially reverted by a probiotic n3-supplemented diet. These results open new possibilities to fight against the noxious effects of obesity.

## Figures and Tables

**Figure 1 antioxidants-13-00161-f001:**
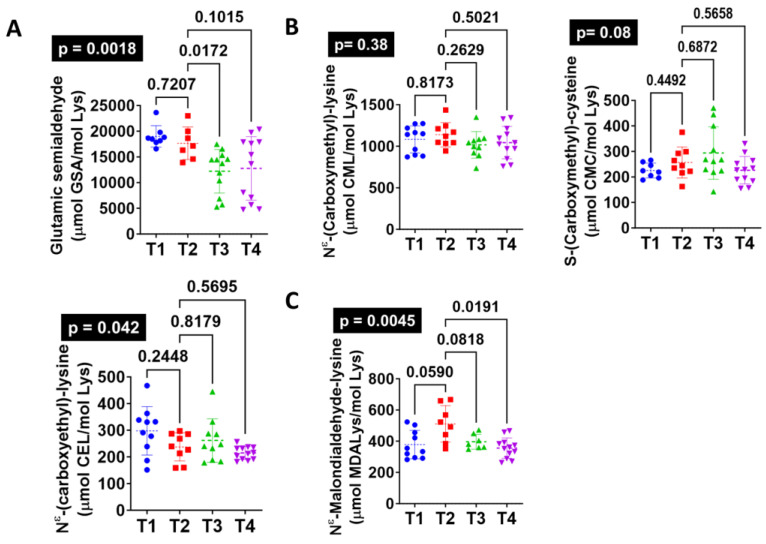
A Western-type diet (T2, *n* = 8) increases lipoxidative damage levels (malondialdehyde-lysine-(MDAL)) in pigs PFC compared with a conventional diet (T1, *n* = 10). These changes can be reverted by supplementing the T2 diet with vegetal protein and *Bifidobacterium breve* CECT8242 alone (T3, *n* = 7) or in combination with n3 fatty acids (T4, *n* = 12). Gas chromatography–mass spectrometry was used to measure protein damage markers of (**A**) direct oxidation: glutamic semialdehyde (GSA); (**B**) glycoxidation: carboxymethyl-lysine (CML), carboxymethyl-cysteine (CMC), and carboxyethyl-lysine (CEL); and (**C**) lipoxidation: malondialdehyde-lysine (MDAL). Global ANOVA *p*-values are highlighted in black boxes.

**Figure 2 antioxidants-13-00161-f002:**
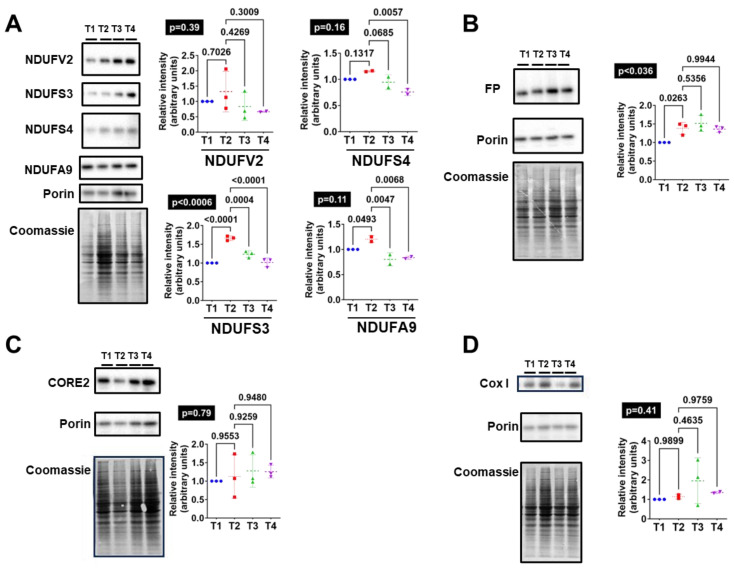
Complex I and II subunits are increased in pigs PFC after feeding a Western-type diet (T2, *n* = 3) compared to a standard one (T1, *n* = 3). These complex I dietary-driven effects are reverted after nutritional supplementation with vegetal protein and *Bifidobacterium breve* CECT8242 alone (T3, *n* = 3) or in combination with n3 fatty acids (T4, *n* = 3). Immunoblots against: (**A**) complex I subunits: NDUFV2, NDUFS3, NDUFS5, and NDUFA9; (**B**) complex II: flavoprotein (FP); (**C**): complex III: CORE2; and (**D**) complex IV: COXI. Porin and Coomassie for each immunoblot is included. Values are expressed as the mean ± SEM and normalized for porin protein levels as a mitochondrial marker. The Western-type diet (T2) increases lipoxidative damage levels (malondialdehyde-lysine-(MDAL)) in pigs PFC compared with a conventional diet (T1). These changes can be reverted by supplementing the T2 diet with vegetal protein and *Bifidobacterium breve* CECT8242 alone (T3) or in combination with n3 fatty acids (T4). Gas chromatography–mass spectrometry was used to measure protein damage markers of (**A**) direct oxidation: glutamic semialdehyde (GSA); (**B**) glycoxidation: carboxymethyl-lysine (CML), carboxymethyl-cysteine (CMC), and carboxyethyl-lysine (CEL); and (**C**) lipoxidation: malondialdehyde-lysine (MDAL). Global ANOVA *p*-values are highlighted in black boxes.

**Figure 3 antioxidants-13-00161-f003:**
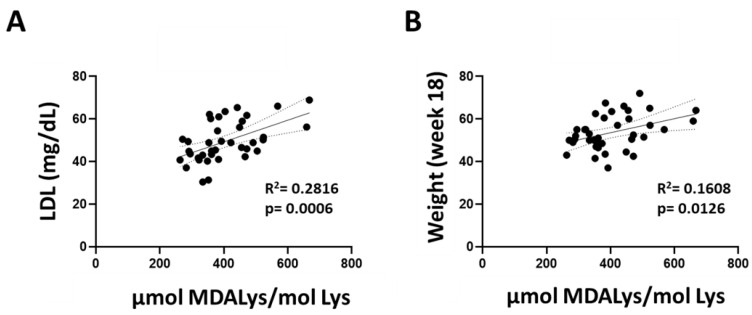
Correlation between protein lipoxidative modification and peripheral traits of lipid metabolism. The degree of the brain cortex lipoxidation is directly proportional to plasma LDL (**A**) levels and animal weight (**B**).

**Table 1 antioxidants-13-00161-t001:** Final body weight and serum lipid measurements.

Variable	Standard (T1) (*n* = 10)	Western-Type (T2) (*n* = 8)	Western-Type + Rice Hydrosilate + *Bifidobacterium* (T3) (*n* = 7)	Western-Type + Rice Hydrosilate + *Bifidobacterium* + n3 Fatty Acids (T4) (*n* = 12)
Body weight (Kg)	53.3 ± 6.4	61.8 ± 7.7 ^T1^	51.36 ± 7.1 ^T2^	50.2 ± 6.8 ^T2^
Triaclyglyceridemia (mM)	0.227 ± 0.056	0.338 ± 0.062 ^T1^	0.377 ± 0.065 ^T1^	0.284 ± 0.082 ^T3^
Total cholesterol (mM)	3.334 ± 0.61	4.219 ± 0.625	4.323 ± 0.823 ^T1^	3.729 ± 0.939
LDL-cholesterol (mM)	1.227 ± 0.232	1.524 ± 0.204 ^T1^	1.322 ± 0.245	1.172 ± 0.197 ^T2^
HDL-cholesterol (mM)	1.169 ± 0.266	1.634 ± 0.391 ^T1^	1.782 ± 0.226 ^T1^	1.528 ± 0.308
LDL/HDL ratio	1.088 ± 0.32	0.948 ± 0.25	0.752 ± 0.16 ^T1^	0.786 ± 0.16 ^T1^

Values are the mean ± SD from a minimum of 8 pigs per group; superindexes indicate a significant difference from the indicated group (*p* < 0.05 of GLM multiple comparisons, post hoc Bonferroni).

**Table 2 antioxidants-13-00161-t002:** Fatty acid composition (mol%, mean ± SEM) and derived indexes and estimated enzyme activities from pigs PFC fed a standard diet (T1, *n* = 10), Western-type diet (T2, *n* = 7), and Western-type diet supplemented with vegetal protein and *Bifidobacterium breve* CECT8242 alone (T3, *n* = 12) or in combination with n3 fatty acids (T4). Gas chromatography was used to measure the content of 21 different FA.

Variable	Standard (T1) *n* = 10	Western-Type (T2) *n* = 8	Western-Type + Rice Hydrosilate + *Bifidobacterium* (T3) *n* = 7	Western-Type + Rice Hydrosilate + *Bifidobacterium* + n3 Fatty Acids (T4) *n* = 12
**Fatty acid**				
C14:0	0.765 ± 0.1	0.831 ± 0.1	0.863 ± 0.1	0.798 ± 0.1
C16:0	24.048 ± 0.3	24.087 ± 0.2	23.453 ± 0.4	23.409 ± 0.3
C16:1 n7	0.784 ± 0.1	0.804 ± 0.1	0.781 ± 0.1	0.803 ± 0.1
C18:0	23.136 ± 0.2	23.268 ± 0.3	23.365 ± 0.2	23.057 ± 0.2
C18:1 n9	18.566 ± 0.5	18.599 ± 0.3	18.871 ± 0.3	18.624 ± 0.4
C18:1 n7	5.629 ± 0.1	5.559 ± 0.1	5.606 ± 0.1	5.314 ± 0.1
C18:2 n6	0.660 ± 0.1	0.723 ± 0.1	0.776 ± 0.1	0.807 ± 0.1
C18:3 n3	0.029 ± 0.01	0.025 ± 0.01	0.026 ± 0.01	0.048 ± 0.01
C18:4 n3	0.122 ± 0.01	0.138 ± 0.01	0.127 ± 0.01	0.119 ± 0.01
C20:0	0.251 ± 0.1	0.250 ± 0.1	0.273 ± 0.1	0.249 ± 0.1
C20:1 n9	0.602 ± 0.1	0.572 ± 0.1	0.594 ± 0.1	0.568 ± 0.1
C20:2 n6	0.032 ± 0.01	0.033 ± 0.01	0.032 ± 0.01	0.029 ± 0.01
C20:3 n6	0.480 ± 0.1	0.518 ± 0.1	0.520 ± 0.1	0.791 ± 0.1
C20:4 n6	9.808 ± 0.2	9.892 ± 0.2	9.431 ± 0.2	9.334 ± 0.1
C20:5 n3	0.012 ± 0.01	0.013 ± 0.01	0.012 ± 0.01	0.022 ± 0.01
C22:0	0.349 ± 0.1	0.352 ± 0.1	0.379 ± 0.1	0.343 ± 0.1
C22:1 n9	0.672 ± 0.1	0.701 ± 0.1	0.705 ± 0.1	0.652 ± 0.1
C22:4 n6	0.099 ± 0.01	0.102 ± 0.01	0.108 ± 0.01	0.088 ± 0.0
C22:5 n6	4.420 ± 0.1	4.442 ± 0.1	4.335 ± 0.1	3.752 ± 0.1
C22:5 n3	0.059 ± 0.01	0.059 ± 0.01	0.061 ± 0.01	0.048 ± 0.011
C22:6 n3	8.969 ± 0.3	9.032 ± 0.3	8.345 ± 0.3	10.838 ± 0.2
**Fatty acid derived index**				
ACL	18.136 ± 0.149	18.282 ± 0.013	17.663 ± 0.429	18.265 ± 0.07
SFA	48.525 ± 0.391	48.789 ± 0.268	48.206 ± 0.531	47.856 ± 0.417
UFA	50.736 ± 0.822	51.211 ± 0.268	48.536 ± 1.902	51.78 ± 0.315
MUFA	26.339 ± 0.649	26.234 ± 0.29	24.767 ± 1.563	25.907 ± 0.424
PUFA	24.397 ± 0.601	24.977 ± 0.375	23.768 ± 0.521	25.873 ± 0.305
PUFAn6	15.383 ± 0.338	15.711 ± 0.213	15.2 ± 0.3	14.799 ± 0.131
PUFAn3	9.014 ± 0.354	9.266 ± 0.269	8.568 ± 0.343	11.074 ± 0.243
DBI	144.168 ± 3.093 ^T2^	146.665 ± 1.835	138.782 ± 3.888 ^T2^	152.387 ± 1.472 ^T2^
PI	139.403 ± 3.898 ^T2^	142.365 ± 2.606	134.382 ± 3.46 ^T2^	150.951 ± 2.26 ^T2^
AI	95.876 ± 3.164	96.797 ± 2.876	94.029 ± 3.261 ^T2^	124.898 ± 2.549 ^T2^
**Estimated elongases and desaturases activity**				
C20:4/C20:3 (Δ5 (n6))	20.603 ± 0.771	19.222 ± 0.6	18.309 ± 0.652	11.932 ± 0.405 ^T2^
C18:4/C18:3 (Δ6 (n3))	4.739 ± 0.47	5.736 ± 0.447	5.027 ± 0.263	2.617 ± 0.266 ^T2^
C20:3/C20:2 (Δ8 (n6))	15.069 ± 0.358	15.935 ± 0.881	16.466 ± 1.26	27.278 ± 1.01 ^T2^
C16:1/C16:0 (Δ9 (n7))	0.033 ± 0.001	0.033 ± 0.001	0.034 ± 0.002	0.034 ± 0.001
C18:1/C18:0 (Δ9 (n9))	1.047 ± 0.026	1.039 ± 0.015	0.979 ± 0.067	1.04 ± 0.023
C20:1/C18:1 (Elovl 3 (n9))	0.025 ± 0.001	0.024 ± 0.001	0.024 ± 0.001	0.024 ± 0.001
C20:2/C18:2 (Elovl 5 (n6))	0.049 ± 0.002	0.047 ± 0.003	0.044 ± 0.003	0.036 ± 0.002
C18:0/C16:0 (Elovl 6 (n9))	0.963 ± 0.014	0.967 ± 0.016	0.998 ± 0.015	0.986 ± 0.009
C20:0/C18:0 (Elovl 1-3-7a (n9))	0.011 ± 0.001	0.011 ± 0.1	0.012 ± 0.001	0.011 ± 0.001
C22:0/C20:0 (Elovl 1-3-7b (n9))	1.381 ± 0.033	1.408 ± 0.024	1.383 ± 0.034	1.37 ± 0.022
C22:4/C20:4 (Elovl 2-5 (n6))	0.01 ± 0.001	0.01 ± 0.001	0.012 ± 0.001	0.009 ± 0.001
C22:5/C20:5 (Elovl 2-5 (n3))	6.253 ± 0.978	5.251 ± 0.519	5.325 ± 0.504	2.22 ± 0.127 ^T2^

Values are mean ± SEM; ACL: average chain length; AI: Antioxidant index; SFA: saturated fatty acids; UFA: unsaturated fatty acids; MUFA: monounsaturated fatty acids; PUFA: polyunsaturated fatty acids; PUFAn6: PUFA n6 series; PUFAn3: PUFA n3 series; DBI: double-bond index; PI: peroxidizability index. T2 Indicate significant differences with T2 group after ANOVA and two-stage linear step-up procedure of Benjamini, Krieger and Yekutieli for controlling false-discovery rate.

## Data Availability

Data are contained within the article and Appendix A.

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
