# Peer review of "High-Fat Diet-Induced Obesity Increases Brain Mitochondrial Complex I and Lipoxidation-Derived Protein Damage"

_antioxidants, 2024, doi:10.3390/antiox13020161_

Round 1

Reviewer 1 Report

Comments and Suggestions for Authors

The study focused on the influence of diet (namely the high-fat diet) on lipids and mitochondrial complexes of the respiratory chain. A pig brain model was used for the study. In the context of neurodegeneration, the influence of diet on brain mitochondria and the link to lipids is of interest in the field. There are already many large studies addressing this topic using WT and disease model-mice (which have the advantage of investigating the relationship to the disease process) . In this study only WT animals were used. There are some questions and comments.

Main comments:

In the abstract and introduction the authors can be more specific about the results. For example, mitochondrial complexes, which ones? and the steady state levels were found to be increased not augmented. Also what does this mean-what is the significance of those findings?

There are no clear descriptions of the statistics and number of animals used to generate the data shown. The information should be described in full in the method section and there should be an explanation of the statistical tests used. The n numbers (animals) should be shown for each graph and table. It is not clear why ~50 animals were used in the study but on some graphs only three data points are shown. What am I looking at in each case? three animals? three independent replications of the experiments?

There is some clarity needed regarding ethics. Please make this a subheading section. If the animals were sacrificed in a research facility (unclear) then was the study approved by the University / institute ethics board? If the animals were sacrificed in an abattoir then the ethics statement should be sufficient. But then the procedure to remove the brain and process the tissue should be stated even if briefly. Were the animals bred, fed and sacrificed in groups? or on different days? Please include the numbers and the study design and whether any animals or samples were pooled or removed from the study.

The figure legends need more detail. Please use p values even if not significant rather than ns. Please check the manuscript for typos, formatting discrepancies, commas in numbers should be a point in English?

Other points and comments:

In the abstract please re-edit the use of the word whose. Who refers to a person not a group of diseases.

  1. Obesity is one of the risk factors for AD, however, as rightfully stated in the limitations the pigs were not aged for the study. I would suggest the authors not call this an AD model but instead, make it a generalized result of high-fat diet-induced obesity to mito damage and lipoxidation protein damage. Additionally, there are other neurological disorders that are associated with obesity.
  2. In Table 1, I would suggest including a ratio of LDL:HDL cholesterol and doing the statistics accordingly. The individual values vary in this study, maybe a ratio here would be more meaningful.
  3. In line 208, instead of the proteome, it would be better if the author just stated the markers of protein oxidation that were tested. The word proteome can be misleading here as one would assume that proteomics was performed.
  4. Line 213, GSA was a more abundant modification in the human brain, maybe the authors can state their hypothesis in the discussion as to why there could be differences in these model systems.
  5. Line 232, instead of 3, 2 subunits have significant differences in Complex 1.
  6. Fig 2, can the authors also quantify if the mitochondrial mass is altered in the T1-T4 conditions
  7. Fig 3, Fig 4 and Table 3, the values of T2, quantification and blot images are repeated in the figures. I suggest to combine the values in Fig 1 and 2 instead. It would be more useful to see if T4 and to some extent, T3 is rescued back to T1 or not.

Comments on the Quality of English Language

Overall good but needs carefully checking for typos, formatting issues and grammar. As mentioned above please resolve any language that can be easily misunderstood. For example the proteome can be misunderstood as proteomics data, over reference to AD when no AD model was used.

Reviewer 2 Report

Comments and Suggestions for Authors

The authors found that feeding female pigs a high fat diet increased protein malondialdehyde levels as a measure of oxidative damage and mitochondrial electron transport chain complex I and II subunit levels. Combining the high fat diet with a probiotic or probiotic plus increased n-3 PUFA were able to abrogate the weight gain from the high fat diet.  The study was rigorous and the data analysis was appropriate. The paper is well-organized and the discussion is strong. However, I have a few minor suggestions to improve the manuscript.

Major comments:

Comment 1: a mitochondrial electron transport chain (ETC) complex I or II activity assay would greatly strengthen the results as it is unknown if increasing the abundance of one or more ETC subunits increases the activity of the entire complex.

Comment 2: Dr. Barja in Madrid and several of this study’s co-authors have found inverse correlations between the levels of complex I subunits NDUFV1, NDUFV2, and NDUFS4 and longevity [ref. 29]. So, increased levels of these subunits likely increase superoxide production leading to lipid peroxidation. Please reference Dr. Pamplona’s, Dr. Jove’s, Dr. Mota-Martorell’s, and Dr. Barja’s review on NDUFV2 and aging [PMID: 33455045] as well as Dr. Pamplona’s, Dr. Jove’s, and Dr. Mota-Martorell’s review on malondialdehyde as an excellent marker of aging [PMID: 33203089]. They are excellent background reading for the reader and the results provided here support the conclusions of those reviews.

Comment 3: Since VDAC1/porin levels are inversely proportional to lifespan when comparing related species, its abundance might be expected to increase under pro-aging conditions such as the high fat diet to facilitate the transport of hydrogen peroxide from the mitochondrial intermembrane space to the cytoplasm. So, therefore it potentially might not be the best loading control for the Western blots. Were any experiments performed to show that VDAC1/porin levels were not altered with the different diets?

Minor comments (wording and grammar):

Line 37: amongst several cells -> in several neural cell types

Line 45: growing -> expanding

Line 68: intermediates -> intermediate

Line 68: proteins' -> protein

Line 110: Line 249: Line 258, Line 270: Line 283: Italicize Bifidobacterium breve

Line 176: (FP) -> (FP) also called SDHA

Line 176: CORE2 -> CORE2 (UQCRC2)

Line 201: treatments -> phenotypes

Line 203: lipemia treats -> serum lipid measurements

Line 207 -> in -> on

Line 208: in -> at

Line 234: subunits -> abundance of the subunits

Line 254: effect -> effect,

Line 267: in -> on

Line 274: in -> on

Line 281: desaturases and elongases -> desaturase and elongase

Line 301: effect -> an effect

Line 303: an -> the

Line 350: regimes -> regimens

Line 353: high -> highly

Line 358: hypocampus -> hippocampus

Line 369: Mitochondria's significance -> The significance of mitochondria

Line 387: including -> including those using

Line 387: modified the -> modify

Line 389: elucidate further -> further elucidate

Line 390: in -> on

Comments on the Quality of English Language

Minor edits requested.

Reviewer 3 Report

Comments and Suggestions for Authors

In this article, the authors examine the effects of a high-fat Western diet in female Duroc pigs compared with a diet enriched with a pro-biotic and finally, a diet with a pro-biotic and n-3 PUFA. The diet containing the pro-biotic and the combination diet improved measures of circulating cholesterol and triglycerides. The model is an underused one in the EU and is highly translational.
However, several clarifications are required and some conclusions appear to be overstated based on the data shown.
Group sizes are not explained sufficiently.
Claims of important effects of high-fat diet on complex I do not appear to be supported by the data presented.

1) It appears that the high-fat Western diet had no great impact upon fatty acid content measures in brain (Table 2) and therefore, the data held within Table 3 are difficult to interpret.
Please place data on fatty acids from all diets within the same table.
Please use statistical comparisons to compare all data, together, for each outcome measure (i.e., please do not separate comparisons of T1 and T2 from comparisons between T2, T3 and T4).
Importantly, please explain how the effects of the pro-biotic and combination diets are to be interpreted and how translationally valuable they are, if the T2 diet is not different from control?

2) Please confirm if ARRIVE guidelines were followed (https://arriveguidelines.org/arrive-guidelines).
For example, how were animals divided among treatment groups - was this done at random?
How were the groups balanced to ensure equal variances within and between groups?

3) Line 127: It is not sufficient to state frontal cortex. Please provide the precise area used for all measurements, in much greater detail.

4) There does not appear to be a statistics section within the methods. What statistics were used? How were comparisons made when data that showed unequal variances?

5) HDL is thought to be protective. What does it mean that all investigative diets increased (or tended to increase) HDL levels (Table 1)?

6) Table 2
Many errors are 0 - has the limit of detection been reached for C18:3 n3, C22:5 n3, C22:0, C20:5 n3, etc?
This will impact data that is currently in Table 3, and also the type of statistical comparison that should be done (if there are unequal variances).

7) For all graphs: Violin plots are not ideal for the small group sizes shown - please use scatter plots with means plus or minus SDs or sems only.

8) The brain tissue was not perfused. What effect does that have on the measures? Will blood content impact the measurements?

9) For all graphs and tables: Please explain clearly how the groups were selected and why the full group was not used (group sizes of 10-11 are mentioned in line 105).
48 animals appear to have started the study (line 100). Please explain what happened to the remaining animals, if 10-11 were used per group (thus 40-44 used).

10) Figure 2A
Two out of 4 graphs suggest no change in complex I.
In another graph there are only two samples in one of the groups (NDUSF4, T2 diet).
Thus, the conclusion that obesity changes complex I does not appear to be based upon strong data.
Please place comparisons within Figure 4 together with Figure 2. Thus, all data examining the same outcome measure (complex I ) should be compared within one statistical comparison and should be shown within one graph.

Minor
Table 1, 2 and 3 - I suggest placing the name of the diet rather than T1, 2, 3 and 4.

Comments on the Quality of English Language

Some small edits are suggested

Line 127 I suggest using "frozen" rather than "freezed"

Line 203, I suggest "treatments" rather than "treats"

Round 2

Reviewer 3 Report

Comments and Suggestions for Authors

The authors have responded in great detail and comprehensively, to all of my critiques and suggestions.

Comments on the Quality of English Language

Very minor issue: Line 280 of the amended manuscript, suggest amending infringed (perhaps "caused" instead).